# Isolation and Absolute Configurations of Diversiform C_17_, C_21_ and C_25_ Terpenoids from the Marine Sponge *Cacospongia* sp.

**DOI:** 10.3390/md17010014

**Published:** 2018-12-28

**Authors:** Xingwang Zhang, Ping-Lin Li, Guo-Fei Qin, Shengying Li, Nicole J. de Voogd, Xu-Li Tang, Guo-Qiang Li

**Affiliations:** 1Key Laboratory of Marine Drugs, Chinese Ministry of Education, School of Medicine and Pharmacy, Ocean University of China, Qingdao 266003, China; zhangxw@qibebt.ac.cn (X.Z.); ipinglin@ouc.edu.cn (P.-L.L.); qinguofei@126.com (G.-F.Q.); 2Shandong Provincial Key Laboratory of Synthetic Biology, CAS Key Laboratory of Biofuels, Qingdao Institute of Bioenergy and Bioprocess Technology, Chinese Academy of Sciences, Qingdao 266101, China; lishengying@qibebt.ac.cn; 3Laboratory of Marine Drugs and Biological Products, National Laboratory for Marine Science and Technology, Qingdao 266235, China; 4Laboratory for Marine Biology and Biotechnology, Qingdao National Laboratory for Marine Science and Technology, Qingdao 266237, China; 5National Museum of Natural History, 2300 RA Leiden, The Netherlands; nicole.devoogd@naturalis.nl; 6College of Chemistry and Chemical Engineering, Ocean University of China, Qingdao 266100, China

**Keywords:** marine sponge, *Cacospongia*, terpenoids

## Abstract

Chemical investigation of MeOH extract of a South China Sea sponge *Cacospongia* sp. yielded 15 terpenoids belonging to three different skeleton-types, including the unusual C_17_
*γ*-lactone norditerpenoids (**1**–**3**), the rare C_21_ pyridine meroterpenoid (**7**), and the notable C_25_ manoalide-type sesterterpenoids (**4**–**6**, **8**–**10**). Compounds **1**–**5** were initially obtained as enantiomers, and were further separated to be optically pure compounds (**1a**, **1b**, **2a**, **2b**, **3a-r**, **3b-r**, **4a**, **4b**, **5a** and **5b**) by chiral HPLC, with a LiAlH_4_ reduction aid for **3**. Compounds **3a**/**3b** (a pair of inseparable enantiomers), **4a**, **5a**, **6**, and **7** were identified as new compounds, while **1a**/**1b** and **2a**/**2b** were obtained from a natural source and were determined for their absolute configurations for the first time. This is also the first time to encounter enantiomers of the well-known manoalide-type sesterterpenoids from nature. The structures with absolute configurations of the new compounds were unambiguously determined by comprehensive methods including HR-ESI-MS and NMR data analysis, optical rotation comparison, experimental and calculated electronic circular dichroism (ECD), and Mo_2_(OAc)_4_ induced circular dichroism (ICD) methods. The cytotoxicity of the isolates against selected human tumor cell lines was evaluated, however, the tested compounds showed no activity against selected cell lines.

## 1. Introduction

Linear terpenoids, represented by linear furano- and pyrrolo-terpenoids, are a unique family of natural products with widespread bioactivities [1,2]. Among all the isolated linear terpenoids, C_25_ linear sesterterpenoids are the major components with more than 200 compounds been isolated, such as nitropyrrolins, heronapyrroles, fukanedones, and manoalide-type sesterterpenoids [3,4,5,6,7]. Inspiringly, the manoalide-type C_25_ sesterterpenoids, which are frequently obtained from marine sponges, have been studied as topical antipsoriatic lead drug candidates for their potent anti-inflammatory activity [8,9]. Apart from the common C_15_ sesquiterpenoids, C_20_ diterpenoids, and C_25_ sesterterpenoids, a few linear terpenoids take irregular C_17_ or C_21_ carbon skeletons [1,2,3,4,5,6,7]. To the best of our knowledge, there are only four C_17_ terpenoids been isolated from red algae *Laurencia viridis* [10], marine spong *Fasciospongia cavernosa* [11], and Chloranthaceae plant *Chloranthus sessilifolius* [12]. Only one of them takes the linear structure. C_21_ linear terpenoids are also a rare class of natural products which are mainly isolated from marine sponges of the genera *Cacospongia, Carteriospongia, Dysidea, Fasciospongia, Hippospongia, Leiosella, Spirastrella*, and *Spongia* [1,2].

Marine sponges of the genus *Cacospongia* (order Dictyoceratida, family Thorectidae) draw much attention for its production of diverse terpenoids [13], such as C_21_-difuran terpenoid cacospongienone A from *C. scalaris* [14], manoalide-type sesterterpenoid cacospongionolide from *C. mollior* [15], and tetracyclic scalarane-type sesterterpenoid scalarin from *C. scalaris* [16]. However, only four out of the total 17 identified *Cacospongia* species have been studied for their chemical constituents. Besides, the unidentified species of *Cacospongia* (*Cacospongia* sp.) with potential morphological challenge are found to be significant resource for novel terpenoids. For example, two new terpenoids, namely cacofuran and (+)-isojaspic acid with novel bridged tricyclic carbon skeletons were isolated from two unidentified *Cacospongia* species collected from Okinawa and Papua New Guinea, respectively [17,18]. Furthermore, the *Cacospongia* derived terpenoids usually exhibit excellent pharmacological potentials, such as antibacterial activity, anti-inflammatory, and cytotoxicity [13,14,15,16,17,18].

In the course of our continuing search for new bioactive metabolites from Xisha Island sponges [19,20], an unidentified *Cacospongia* species was collected and investigated for its chemical components, yielding three pairs of rare C_17_
*γ*-lactone norditerpenoid enantiomers (**1a**, **1b**, **2a**, **2b** and the inseparable **3a**/**3b**), an infrequent C_21_ pyridine terpenoid (**7**), and six notable C_25_ manoalide-type sesterterpenoids (**4**–**6**, **8**–**10**) including two pairs of enantiomers (**4a**, **4b**, **5a** and **5b**) (Figure 1). Among them, (±)-8,13-secoepicavernosine (**3a**/**3b**), (+)-hippolide E (**4a**), (+)-(6*E*)-neomanoalide (**5a**), (3*R*,4*R*)-14,18-secoluffariolide C (**6**), and cacospongine A (**7**) were identified as new compounds, while the enantiomers **1a**/**1b** and **2a**/**2b** were separated by chiral HPLC. This is the first time to obtain manoalide-type enantiomers from nature. The structures with absolute configurations of the new compounds were unambiguously elucidated by a combinatorial methods, including HR-ESI-MS, 1D and 2D NMR spectra analysis, optical rotation comparison, experimental and calculated ECD comparison, and Mo_2_(OAc)_4_ induced circular dichroism (ICD) method. Furthermore, cytotoxicity of these compounds was tested against selected tumor cell lines. Herein, we report the isolation, chiral resolution, structure elucidation, and bioactivity of the isolated linear terpenoids.

## 2. Results and Discussion

Compound **1** was obtained as colorless oil, and its molecular formula was determined as C_17_H_28_O_3_ by HR-ESI-MS (*m*/*z* 303.1936, [M+Na]^+^, calcd 303.1931, Appendix A). A comparison of the ^1^H and ^13^C NMR data (Table 1) with literature suggested **1** to be the reported cavernosine definitely [11]. However, compound **1** showed neither optical activity (${\left[\alpha \right]}_{D}^{20}\sim 0$) nor Cotton effect on its CD spectra, just as cavernosine did in the original report (${\left[\alpha \right]}_{D}^{20}-1.8$) [11], which suggested **1** to be a racemic mixture. Then, chiral HPLC resolution of **1** afforded a pair of stereoisomers **1a** (${\left[\alpha \right]}_{D}^{20}+7.5$) and **1b** (${\left[\alpha \right]}_{D}^{20}-10.1$) with a ratio of approximately 1:1 (Appendix A). The relative configuration of the C-4–C-5 fragment in **1a** and **1b** were both assigned as *erythro* type (4*S**,5*R**) by comparing the ^1^H and ^13^C NMR data (Table 1, Appendix A) with those of the synthesized *erythro* (±)-cavernosine and *threo* (±)-epicavernosine [21]. The absolute configurations of **1a** and **1b** were established by ECD methods. According to literature, the *γ*-chiral center of a saturated *γ* lactone would induce a Cotton effect at 213 nm, which is mainly attributed to the *π_x_*→*π_x_** electronic transition of the carbonyl group in the *γ*-lactone ring [22]. Accordingly, **1a** and **1b** were found to take a negative and positive Cotton effects at 213 nm, respectively. Thus, **1a** and **1b** were proposed to have the 4*S* and 4*R* absolute configurations [22]. The result was further confirmed by a comparison of the experimental ECD spectra with the calculated curves of the four possible candidate stereoisomers of **1**. The calculated ECD data suggested that the valuence of the Cotton effect at 213 nm was solely related to the absolute configuration of C-4 (negative-4*S*, positive-4*R*) (Figure 2), while the Cotton effect intensity could be affected by the auxochromic hydroxy group at C-5. Finally, the absolute configurations of **1a** [(+)-cavernosine] and **1b** [(−)-cavernosine] were assigned as 4*S*,5*R* and 4*R*,5*S*, respectively. This is the first time to separate enantiomers of cavernosine and to determine their absolute configurations.

Compounds **2** and **3**, having the same molecular formula of C_17_H_28_O_3_ with **1**, were isolated as colorless oil. A comparison of the 1D NMR data of **2** with those of **1** (Table 1, Appendix A) suggested that they both take the same *γ*-lactone ring and C-4-C-7 fragments. However, the rest fragment of **2** is obviously different with that of **1** according to their 1D data, especially the two methine signals (*δ*_H_ 5.08, t, *J* = 7.1 Hz and *δ*_H_ 5.04, t, *J* = 6.9 Hz) observed in **2**. HMBC correlations from H_3_-14/H_3_-15 to two alkenyl carbon signals at *δ*_C_ 124.0 (C-12) and *δ*_C_ 131.2 (C-13) strongly suggested the formation of a double bound of C-12 = C-13 in **2** (Appendix A). Thus compound **2** was determined as a 8,13-seco product of **1**. Compound **2** has been synthesized as a side product [23]. The mostly identical 1D and 2D NMR data of **3** and **2** indicated that they both take the same planar structure. The slight difference of Me-17 in **2** and **3** (*δ*_H_ 1.14, *δ*_C_ 20.9 in **3** vs. *δ*_H_ 1.26, *δ*_C_ 23.0 in **2**) suggested they may be a pair of epimers. Geometrical configurations of double bonds ∆^8^ in **2** and **3** were both deduced as *E* geometry, which was deduced from NOE correlations of H_3_-16/H_2_-7 and H_2_-10/H-8, and from the chemical shifts of C-16 (<20 ppm, a methyl resonance appearing at a chemical shift less than 20 ppm is indicative of an ‘*E*’ configuration, whereas a value larger than 20 ppm indicates a ‘*Z*’ configuration) [24]. Comparisons of chemical shifts of H_3_-17 in **2** (*δ*_H_ 1.02; C_6_D_6_) and **3** (*δ*_H_ 0.74; C_6_D_6_) with that of the synthesized *erythro* 8,13-secocavernosine (*δ*_H_ 1.08; C_6_D_6_) revealed **2** possessing a *erythro*-type relative configuration for the C-4–C-5 fragment [23]. While chemical shifts of Me-17 in **3** (*δ*_H_ 1.14 vs. *δ*_H_ 1.26 in **2**; CDCl_3_) was in agreement with that of the synthesized *threo*-type analog [(*S*)-5-((*S*)-2-hydroxy-6-methylhept-5-en-2-yl)-dihydrofuran-2(3H)-one] (*δ*_H_ 1.16; CDCl_3_) [25]. Thus, the relative configurations of **2** and **3** were assigned as *erythro* (4*S**,5*R**) and *threo* (4*S**,5*S**), respectively. Optical rotation value and ECD experiment disclosed that **2** and **3** were both optical inactive. Further chiral resolution of **2** afforded a pair of stereoisomers **2a** (${\left[\alpha \right]}_{D}^{20}+6.5$) and **2b** (${\left[\alpha \right]}_{D}^{20}-10.4$) (Appendix A). The absolute configurations of **2a** and **2b** were established as 4*S*,5*R* and 4*R*,5*S*, respectively, by experimental and calculated ECD spectra comparison (Figure 3) Thus, **2a** and **2b** were finally established as (+)-8,13-secocavernosine and (−)-8,13-secocavernosine, respectively.

However, the optical inactive compound **3** could not be further separated by chiral HPLC (Appendix A). To address the absolute configuration of **3**, several chemical tailoring reactions were tried. During the NaOH hydrolysis of **3**, the hydrolytic product **3-h** was unstable and spontaneously formed the original *γ*-lactone **3**, even using CH_2_N_2_ as a protective agent for the carboxyl of **3-h** (Appendix A). Finally, compound **3** was successfully reduced to acyclic 1,4,5-triol derivative using LiAlH_4_ (Scheme 1) [11]. The reduced product was further separated to be a pair of isomers **3a-r** (${\left[\alpha \right]}_{D}^{20}-7.8$) and **3b-r** (${\left[\alpha \right]}_{D}^{20}+11.1$), with a ratio of 1:1 on chiral HPLC (Scheme 1). On the basis of Snatzke’s theory [26,27], Mo_2_(OAc)_4_ induced circular dichroism (ICD) spectra of acyclic adjacent diol substrate (relative configuration determined) could be used to established the absolute configurations of the diol fragment according to the key diagnostic Cotton effect at 310 nm (band IV). Thus, Mo_2_(OAc)_4_ ICD experiments was carried out, and a positive and a negative Cotton effect at 310 nm on the ICD spectra of **3a-r** and **3b-r** were observed, respectively. The absolute configurations of **3a-r** and **3b-r** were determined to be 4*S*,5*S* and 4*R*,5*R* (Figure 4), respectively [26,27]. This result confirmed **3** to be a pair of inseparable new enantiomers bearing the 4*S*,5*S* and 4*R*,5*R* absolute configurations.

Compounds **4a**/**4b** and **5a**/**5b**, having the same molecular formula of C_25_H_40_O_4_ as deduced by HR-ESI-MS data (Appendix A), were obtained as two pairs of enantiomers both with an approximate ratio of 1:3 (Appendix A). Each pair of isomers possess the same NMR data (Table 2), while their optical rotations (${\left[\alpha \right]}_{D}^{20}$
**4a** + 4.1 vs. **4b** − 7.4; **5a** + 3.6 vs. **5b** − 7.7) and ECD curves (Figure 5) are almost opposite. Comparisons of 1D NMR data of **4a**/**4b** and **5a**/**5b** (Table 2, Appendix A) with those of literatures revealed their planar structure to be hippolide E and (*E*)-neomanoalide, respectively [28,29]. The absolute configurations of **4b** [(−)-hippolide E] and **5b** [(−)-(*E*)-neomanoalide] were assigned to be the same as hippolide E and (6*E*)-neomanoalide-24-ol [28,29] by CD spectra comparisons. Their enantiomers **4a** [(+)-hippolide E] and **5a** [(+)-(6*E*)-neomanoalide-24-ol] were both deduced to be 4*S* configuration according to their opposite CD absorptions (Figure 5) and opposite optical rotations comparing with **4b** and **5b**. Thus, **4a** and **5a** were identified as two new enantiomers of hippolide E and (*E*)-neomanoalide-24-ol, respectively. This is the first time to obtain the natural-occurring enantiomers of the well-known manoalide-type sesterterpenoids.

Compound **6**, with the molecular formula of C_25_H_40_O_4_, was also considered as a manoalide-type sesterterpenoid according to its similar 1D NMR data with those of **4a**/**4b** and **5a**/**5b** (Table 2, Appendix A). Further 1D NMR data comparison of **6** with those of co-isolated luffariolide C (**8**) revealed their structural similarity, except for a slight difference of Me-20 (*δ*_H_ 1.68 in **6** vs. *δ*_H_ 0.99 in luffariolide C) and Me-21 (1.60 vs. 0.99) [30]. HMBC correlations from H_3_-20/H_3_-21 to the olefinic carbons of C-18 (*δ*_C_ 124.3) and C-19 (*δ*_C_ 131.3) (Figure 6, Appendix A) suggested that the cyclohexene ring in luffariolide C was cleaved and formed an extra double bound of C-18=C-19 in **6**. Further 2D NMR analysis, such as ^1^H-^1^H COSY correlations of H_2_-16/H_2_-17/H-18 and HMBC correlations from H_3_-22 to C-14/C-15/C-16, (Figure 6) confirmed the planar structure of **6** to be 14,18-seco luffariolide C. Double bounds of ∆^10^ and ∆^14^ were both deduced to be *E* geometry according to NOE correlations of H_3_-23/H_2_-9 and H_3_-22/H_2_-13 (Figure 7). While double bound of ∆^6^ was assigned as *Z* geometry by NOE correlation of H_2_-24/H_2_-5. Additionally, NOE correlations of H-3/H_2_-5 and H-4/H_2_-25 (Figure 7), as well as the coupling constant between H-3 and H-4 (*J* = 0 Hz) in **6**, suggested it takes the same 3,4-*trans* relative configuration with luffariolide C. Moreover, the similar optical rotations of **6** (${\left[\alpha \right]}_{D}^{20}=+3.9$) and luffariolide C (${\left[\alpha \right]}_{D}^{20}=+4.4$) revealed the same 3*R*,4*R* absolute configurations. The result was further supported by experimental and theoretical ECD comparisons of the main chiral lactone structure in **6** (Figure 8). Thus, **6** was determined as (3*R*,4*R*)-14,18-secoluffariolide C.

The molecular formula of compound **7** was deduced as C_21_H_31_N by HR-ESI-MS at *m*/*z* 298.2532 ([M + H]^+^ calcd 298.2529, Appendix A). ^1^H NMR spectrum of **7** showed four aromatic proton signals (*δ*_H_ 8.40, s; 8.38, d, *J* = 4.1 Hz; 7.60, d, *J* = 7.6 Hz; 7.27, dd, *J* = 6.7, 5.6 Hz). The remaining proton signals in upfield were similar with those of **6**, especially the four olefinic methyls of H_3_-18 (*δ*_H_ 1.63 in **7** vs. 1.68 in **6**), H_3_-19 (1.54 vs. 1.60), H_3_-20 (1.54 vs. 1.60) and H_3_-21 (1.47 vs. 1.60). ^1^H-^1^H COSY correlations of H_2_-6/H_2_-7/H-8, H_2_-10/H_2_-11/H-12 and H_2_-14/H_2_-15/H-16, together with HMBC correlations from H_3_-18/ H_3_-19 to C-16/C-17, from H_3_-20 to C-12/C-13/C-14, and from H_3_-21 to C-8/C-9/C-10, indicated a C_16_ prenyl chain (C-6-C-18). Additionally, a monosubstituted pyridine ring was constructed based on ^1^H-^1^H COSY correlations of H-3/H-4/H-5 and HMBC correlations from H-5 to C-1/C-3, from H-3 to C-2 and from H-1 to C-5 (Figure 6, Appendix A). The C_16_ linear chain was connected with the pyridine ring at C-2 by HMBC correlations from H_2_-6 (*δ*_H_ 2.60, t, *J* = 7.6 Hz) to C-1 (*δ*_C_ 149.6), C-2 (*δ*_C_ 137.0) and C-3 (*δ*_C_ 135.8) (Figure 6). The *E* geometry of double bounds ∆^8^ and ∆^12^ were both deduced from the chemical shifts of C-21 and C-20 (*δ*_C_ both at 15.7, < 20 ppm) [24]. Thus compound **7** was determined as a C_21_ pyridine terpenoid and was named as cacospongine A.

The structures of the known analogs of luffariolide C (**8**) [30], (*Z*)-neomanoalide (**9**) [29] and hippolide J (**10**) [31] were determined by comparison of their 1D/2D NMR data, ESI-MS data and optical rotations with those of reported values.

The cytotoxicity of the isolates against human tumor cell lines of K562, HCT116, Hep3B, A-549, and Jurkat was evaluated, using MTT method [32] with adriamycin as positive control. However, all the tested compounds were inactive (IC_50_ > 10 μM).

## 3. Experimental Section

### 3.1. General Methods

Optical rotations were measured on a JASCO P-1020 digital polarimeter (JASCO Corporation, Tokyo, Japan). UV spectra were measured on a Beckman DU640 spectrophotometer. ECD spectra were obtained on a Jasco J-815 CD spectrometer. IR spectra were recorded on a Nicolet Nexus 470 (FT-IR) spectrophotometer (Thermo Electron Co., Madison, WI, USA), KBr pellets. NMR spectra were recorded on a Bruker DRX-500MHz instrument (Bruker BioSpin GmbH Co., Rheinstetten, Germany), 500 MHz for ^1^H NMR and 125 MHz for ^13^C NMR in CDCl_3;_ chemical shifts *δ* in ppm referred to the solvent peaks at δ_H_ 7.26 and δ_C_ 77.0 for CDCl_3_, δ_H_ 2.50 and _C_ 39.5 for DMSO-d_6_ and δ_H_ 7.16 for C_6_D_6_, and coupling constant *J* in Hz. HR-ESI-MS were obtained from a Micromass Q-Tof Ultima GLOBAL GAA076 LC-mass spectrometer (Waters Corporation, Milford, MA, USA). HPLC separation was performed on an Agilent 1100 series instrument with DAD detector (Agilent technologies, Santa Clara, CA, USA), equipped with a semi-preparative reversed-phased column (YMC-packed C18, 5 μm, 250 × 10 mm, 1.5 mL/min) or an analytic chiral column DAICEL IC-3 (DAICEL chiral technologies, Shanghai, China). Precoated silica gel plates (GF_254_, Qingdao Marine Chemical Inc., Qingdao, China) were used for TLC analyses. Silica gel (200–300 mesh, Qingdao Marine Chemical Inc., Qingdao, China) was used for column chromatography (CC).

### 3.2. Animal Material

The sponge *Cacospongia* sp. is in irregular shape, with black-brownish color, and covered with a tight encrusting crust. There are irregularity distributed holes on the surface, wihch is thick and solid, and is in the dimensions of around 4 × 5 cm ~ 6 × 8 cm (see Appendix A. Photos of sponge specimen). The specimen was collected from the coral-reef regions of Yong Xing Island (16°50′ N, 112°20′ E) in the South China Sea, at a depth ranging from 18 to 25 m, in November 2010. The specimen, with a voucher of no. XS-2009-34, was frozen immediately at −20 °C until it was examined. The voucher was deposited at the School of Medicine and Pharmacy, Ocean University of China, China. The sponge was identified by Dr. N.J.d.V. (Naturalis Biodiversity Centre, Leiden, The Netherlands).

### 3.3. Extraction and Isolation

The frozen sponge (2.6 kg, wet weight) was minced and extracted with MeOH for three times (each time for one day) at room temperature (5 L × 3). The combined solution was evaporated in vacuum and desalinated for three times to yield a residue (56.0 g). The crude extract was then subjected to a reduced pressure silica gel column eluting with a step-by-step gradient elution of acetone-petroleum ether (from 0:1 to 1:0, *v*/*v*), to give six fractions. Fraction 2 was then further separated by silica gel column eluting with petroleum ether/acetone (2:1, 1:1, 0:1, *v*/*v*) to give subfractions F2-1–F2-5. F2-2 was further purified by semi-preparative HPLC with a mobile phase of MeOH/H_2_O (70:30, *v*/*v*) to give three pairs of isomer mixtures **1** (6.0 mg; *t*_R_ 53.0 min), **2** (30.1 mg; *t*_R_ 59.0 min) and **3** (4.1 mg; *t*_R_ 56.0 min). These isomers were finally separated by chiral HPLC with a mobile phase of n-hexane/ isopropanol (93:7, *v*/*v*), and yielded **1a** (2.3 mg; *t*_R_ 15.0 min), **1b** (2.4mg; *t*_R_ 28.0 min), **2a** (12.1 mg; *t*_R_ 15.0 min), **2b** (12.0 mg; *t*_R_ 24.5 min), **3a/3b** (enantiomeric mixture, 4.1 mg; *t*_R_ 17.5 min). HPLC purification (85:15, MeOH/H_2_O, *v*/*v*) of F2-3 afforded compound **7** (6.1 mg; *t*_R_ 37.0 min). Fraction 3 was chromatographed on another silica gel column to give two subfractions F3-1 and F3-2, which were found to contain terpenoid components by TLC analyses. Then F3-1 was further separated by HPLC (85:15, MeOH/H_2_O, *v*/*v*) to give **6** (2.4 mg; *t*_R_ 42.0 min), **8** (1.5 mg; *t*_R_ 37.0 min), **9** (4.6 mg; *t*_R_ 56.0 min) and **10** (5.0 mg; *t*_R_ 61.0 min). The two pairs of isomers of **4a** (1.6 mg; *t*_R_ 47.0 min)/**4b** (4.2 mg; *t*_R_ 42.0 min) and **5a** (1.5 mg; *t*_R_ 49.0 min) /and **5b** (4.3 mg; *t*_R_ 44.0 min) were ultimately separated by chiral HPLC with a mobile phase of n-hexane/ethanol (94:6, *v*/*v*) from F3-2.

*Cavernosine* (**1**)*:* colorless oil; (+)*-cavernosine* (**1a**), ${\left[\alpha \right]}_{D}^{20}+7.5$ (*c* 0.1, MeOH); *(−)-cavernosine* (**1b**), ${\left[\alpha \right]}_{D}^{20}-10.1$ (*c* 0.1, MeOH); IR (KBr) *v*_max_ 3479, 2926, 1779, 1450, 1374, 1191 cm^−1^; ^1^H NMR (CDCl_3_, 500 MHz) and ^13^C NMR (CDCl_3_, 125 MHz), see Table 1; HRESIMS *m*/*z* 303.1936 [M + Na]^+^ (calcd for C_17_H_28_O_3_Na, 303.1931).

*8,13-secocavernosine* (**2**): colorless oil; *(+)-8,13-secocavernosine* (**2a**)*:*
${\left[\alpha \right]}_{D}^{20}=+3.9$ (*c* 0.1, MeOH); *(−)-8,13-secocavernosine* (**2b**), ${\left[\alpha \right]}_{D}^{20}-10.4$ (*c* 0.1, MeOH); IR (KBr) *v*_max_ 3469, 2928, 1774, 1453, 1377, 1191 cm^−1^; ^1^H NMR (CDCl_3_, 500 MHz) and ^13^C NMR (CDCl_3_, 125 MHz), see Table 1; ^1^H NMR in C_6_D_6_ (500 MHz) *δ* 5.22 (t, *J* = 6.9 Hz, 1H), 5.16 (t, *J* = 6.9 Hz, 1H), 3.64 (t, *J* = 7.5 Hz, 1H), 1.68 (s, 3H), 1.57 (s, 3H), 1.57 (s, 3H), 1.02 (s, 3H); HRESIMS *m*/*z* 303.1938 [M + Na]^+^ (calcd for C_17_H_28_O_3_Na, 303.1931).

*(±)-8,13-secoepicavernosine* (**3a/3b**, unseparated)*:* colorless oil (MeOH); ${\left[\alpha \right]}_{D}^{20}\sim 0$ (*c* 0.1, MeOH); IR (KBr) *v*_max_ 3473, 2919, 1774, 1649, 1456, 1379, 1190, 1004 cm^−1^; ^1^H NMR (CDCl_3_, 500 MHz) and ^13^C NMR (CDCl_3_, 125 MHz), see Table 1; ^1^H NMR in C_6_D_6_ (500 MHz) *δ* 5.25 (t, *J* = 6.9 Hz, 1H)), 5.20 (t, *J* = 7.0 Hz, 1H), 3.62 (t, *J* = 7.5 Hz, 1H), 1.70 (s, 3H), 1.61 (s, 3H), 1.59 (s, 3H), 0.74 (s, 3H); HRESIMS *m*/*z* 303.1936 [M + Na]^+^ (calcd for C_17_H_28_O_3_Na, 303.1931).

*(+)-hippolide E* (**4a**)*:* colorless oil (MeOH); ${\left[\alpha \right]}_{D}^{20}=+4.1$ (*c* 0.1, MeOH); IR (KBr) *v*_max_ 3410, 2925, 2360, 2338, 1746, 1700, 1650, 1540, 1455, 1381 cm^−1^; ^1^H NMR (CDCl_3_, 500 MHz) and ^13^C NMR (CDCl_3_, 125 MHz), see Table 2; HRESIMS *m*/*z* 425.2671 [M + Na]^+^ (calcd for C_25_H_38_O_4_Na, 425.2662).

*(+)-(6E)-neomanoalide* (**5a**)*:* colorless oil (MeOH); ${\left[\alpha \right]}_{D}^{20}=+3.6$ (*c* 0.1, MeOH); IR (KBr) *v*_max_ 3415, 2926, 2860, 2359, 2338, 1744, 1647, 1455, 1380, 1143, 1063 cm^−1^; ^1^H NMR (CDCl_3_, 500 MHz) and ^13^C NMR (CDCl_3_, 125 MHz), see Table 2; HRESIMS *m*/*z* 425.2674 [M + Na]^+^ (calcd for C_25_H_38_O_4_Na, 425.2662).

*(3R,4R)-14,18-secoluffariolide C* (**6**)*:* colorless oil (MeOH); ${\left[\alpha \right]}_{D}^{20}=+3.9$ (*c* 0.1, MeOH); IR (KBr) *v*_max_ 3384, 2922, 2856, 2359, 2337, 1750, 1451, 1375, 1265, 1196, 1018 cm^−1^; ^1^H NMR (CDCl_3_, 500 MHz) and ^13^C NMR (CDCl_3_, 125 MHz), see Table 2; HRESIMS *m*/*z* 405.2995 [M + H]^+^ (calcd for C_25_H_41_O_4_, 405.2999).

*Cacospongine A* (**7**)*:* colorless oil (MeOH); IR (KBr) *v*_max_ 2924, 2854, 2361, 2339, 1718, 1453, 1379, 1190 cm^−1^; ^1^H NMR (DMSO-*d*_6_, 500 MHz) and ^13^C NMR (DMSO-*d*_6_, 125 MHz), see Table 2; HRESIMS *m*/*z* 298.2532 [M + H]^+^ (calcd for C_21_H_32_N, 298.2529).

### 3.4. LiAlH_4_ Reduction of ***3***

The solution of **3** (3 mg) in dry THF (2 mL) containing LiAlH_4_ (3 mg) were stirred for 6 h at room temperature, under the protection of argon atmosphere. The reduction reaction was quenched by addition of 2 mL of 10% KOH aqueous solution, and the mixture was then extracted by EtOAc to yielded crude products (2.7 mg) [11]. The isomer mixtures were further separated by chiral HPLC (n-hexane/ isopropanol, 93:7, *v*/*v*) to give **3a-r** (1.3 mg; *t*_R_ 11.7 min) and **3b-r** (1.2 mg; *t*_R_ 12.7 min).

*Compounds***3a-r** and **3b-r**: colorless oil from MeOH; ${\left[\alpha \right]}_{D}^{20}-7.8$ for **3a-r** and +11.1 for **3b-r** (*c* 0.1, MeOH); ^1^H NMR (CDCl_3_, 500 MHz) *δ* 5.14 (1H, t, *J* = 6.7 Hz, H-8), 5.08 (1H, t, *J* = 6.7 Hz, H-12), 3.70 (2H, m, H_2_-1), 3.46 (1H, d, *J* = 10.9 Hz, H-4), 2.11 (1H, m, H-7a), 2.06 (1H, m, H-7b), 2.06 (2H, m, H_2_-11), 1.98 (2H, m, H_2_-10), 1.75 (2H, m, H_2_-2), 1.68 (3H, s, H_3_-14), 1.62 (3H, s, H_3_-16), 1.60 (3H, s, H_3_-15), 1.56 (1H, m, H-6a), 1.50 (each 1H, m, H-6b, H-3a), 1.40 (1H, m, H_2_-3b), 1.12 (3H, s, H_3_-17); ^13^C NMR (CDCl_3_, 125 MHz) *δ* 135.8 (C, C-9), 131.5 (C, C-13), 124.2 (CH, C-12), 124.1 (CH, C-8), 77.2 (CH, C-4), 75.1 (C, C-5), 62.9 (CH_2_, C-1), 39.7 (CH_2_, C-10), 38.8 (CH_2_, C-6), 30.1 (CH_2_, C-2), 28.6 (CH_2_, C-3), 26.6 (CH_2_, C-11), 25.7 (CH_3_, C-14), 20.8 (CH_3_, C-17), 22.0 (CH_2_, C-7), 17.7 (CH_3_, C-15), 16.0 (CH_3_, C-16).

### 3.5. Determination of the Absolute Configuration of the Diol Moiety in ***3a-r*** and ***3b-r*** by Snatzke’s Method

ICD spectra of Mo-complexes of **3a-r** and **3b-r** were obtained according to reported procedures [26,27]. The reduced products (each 0.5 mg) and Mo_2_(OAc)_4_ (1.0 mg) were dissolved in 1.5 mL of dry DMSO to give a solution, with the ligand to metal molar ratio being around 1:1.2. The electronic transitions of the metal complexes in DMSO were monitored by a Jasco J-815 CD spectrometer in the UV–vis region of 250–500 nm, and stationary ICD spectra were obtained after 50 min at 15 °C. Because there were no inherent absorptions for the reduced products, the observed ICD spectra could be directly used to analyze the absolute configurations of diol fragments in the ligands with the characteristic cotton effects around 310 nm, according to Snatzke’s theory [26,27].

### 3.6. Calculating Section

The quantum chemical calculations were performed using the density functional theory (DFT) by Gaussian 09 [33]. The initial key chiral structures in compounds **1a**, **1b**, **2a**, **2b** and **6** were built with Spartan 10 software, and all trial structures were first minimized based on molecular mechanics calculations. Conformational search was performed by Spartan 10 software using MMFF force filed, and conformers occurring within a 10 kcal/mol energy window from the global minimum were chosen for geometry optimization in the gas phase with the DFT method at the B3LYP/DGDZVP level. The B3LYP/DGDZVP harmonic vibrational frequencies were further calculated to confirm their stability. The spin-allowed excitation energies and rotatory (*R*n) and oscillator strengths (*f*n) of the lowest excited states of stable conformers were calculated for ECD spectra using TD-DFT method with the basis set RB3LYP/DGDZVP. Solvent effects of methanol solution were evaluated at the same DFT level by using the SCRF/PCM method in agreement with the experiment condition. Electronic transitions were expanded as Gaussian curves with a FQHM (full width at half maximum) for each peak of 0.32 eV. The ECD spectra were combined after Boltzmann weighting according to their population contribution.

### 3.7. Cytotoxicity

In vitro cytotoxicity was determined by MTT [3-(4,5-dimethylthiazol-2-yl)-2,5-diphenyltetrazolium bromide] colorimetric assay against K562, HCT116, Hep3B, A-549 and Jurkat cell lines by reported procedures [32]. All the cell lines were purchased from Shanghai Institute of Cell Biology (Shanghai, China). Adriamycin (doxorubicin, ADM) was used as a positive control.

## 4. Conclusions

In summary, the present study of sponge *Cacospongia* sp. led to the identification of 15 optically pure terpenoids. This is the first time to encounter a series of stereoisomers of the rare linear C_17_ terpenoid and well-known manoalide-type sesterterpenes from an individual sponge species. Furthermore, there is only one C_17_
*γ*-lactone terpenoid named cavernosine and no C_21_ pyridine terpenoid had been reported before [10]. The structural diversity of manoalides was usually derived from the multi-site cyclization andoxidation of the linear prenyl chain, such as compounds **5**, **8**–**10**. However, two pairs of enantiotopic manoalide-type sesterterpenoids (**4a**/**4b** and **5a**/**5b**) expanded the chemical diversity of manoalides by a stereoisomer manner. The present results showed a remarkable structural diversity of terpenoids in the sponge *Cacospongia* sp. The unique C_17_ and C_21_ meroterpenoids showed chemotaxonomy significance for the unidentified species of *Cacospongia*.

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
