# Peer review of "Isolation and Absolute Configurations of Diversiform C17, C21 and C25 Terpenoids from the Marine Sponge Cacospongia sp."

_marinedrugs, 2018, doi:10.3390/md17010014_

Round 1

Reviewer 1 Report

This paper describes the isolation and absolute configurations of diversiform C17, C21 and C25 terpenoids from marine sponge cacospongia sp. I could recommend the manuscript for publication subject to some revisions as indicated below:

1. Lines 38-39.

“… Mo2(OAc)4 induced circular dichroism (ICD) after chemical tailoring. The cytotoxicity of the isolates against selected human tumor cell lines was evaluated”.

Since the tested compounds were inactive with IC50 > 10 μΜ, this should be denoted in the abstract. In fact, this a major drawback of the manuscript.

2. Line 135.

“Figure 3. Key COSY (▬) and HMBC (→) correlations of 2a/2b, 3, 6 and 7.”.

The notation of the atoms should be denoted.

3. Lines 241-242.

The frozen sponge (2.6 kg, wet weight) was minced and extracted with MeOH for three times at room temperature (5L x 3).”.

The extraction time should be included.

4. Tables S1 and S3.

The thermodynamic parameters should be expressed not only in a.u. units but also in kcal/mol.

5. Figures S9, S10, S11, S13, S14, S15 etc.

The assignment of the most prominent peaks should be provided.

6. Figures S39 and S47

It is not clear the anti-phase character of the HSQC cross peaks.

Author Response

1. Lines 38-39.

“… Mo2(OAc)4 induced circular dichroism (ICD) after chemical tailoring. The cytotoxicity of the isolates against selected human tumor cell lines was evaluated”.

Since the tested compounds were inactive with IC50 > 10 μΜ, this should be denoted in the abstract. In fact, this a major drawback of the manuscript.

The negative result of cytotoxic test has been added into the abstract part.

2. Line 135.

“Figure 3. Key COSY (▬) and HMBC (→) correlations of 2a/2b, 3, 6 and 7.”.

The notation of the atoms should be denoted.

The notation of the atoms has been denoted in Figure 3.

3. Lines 241-242.

The frozen sponge (2.6 kg, wet weight) was minced and extracted with MeOH for three times at room temperature (5L x 3).”.

The extraction time should be included.

The extraction time has been add into this part.

4. Tables S1 and S3.

The thermodynamic parameters should be expressed not only in a.u. units but also in kcal/mol.

We have added kcal/mol as thermodynamic parameters in this part.

5. Figures S9, S10, S11, S13, S14, S15 etc.

The assignment of the most prominent peaks should be provided.

We have provided the most prominent peaks in all of the 1D NMR spectra.

6. Figures S39 and S47

It is not clear the anti-phase character of the HSQC cross peaks.

We have optimized these two HSQC spectra quality.

Reviewer 2 Report

The manuscript by Zhang et al describes the isolation and structure determination of a number of terpenoids from Cacospongia sp. From my point of view, the work is in general technically well done. There is an exception in compound 6. With regard to the configurations of the cyclic moiety, the arguments given are not enough. It has beed reported that NOE between vicinal protons in five member rings are inconclusive. 1-3 NOEs or heteronuclear coupling constants should be used instead (Napolitano et al Chem. Eur. J. 2011, 17, 6338 – 6347 and Constantino et al J. Org. Chem. 2008, 73, 6158–6165.). However, I have to say that from my point of view, the work is not interesting enough to be published in a Q1 journal such as Marine Drugs.

Author Response

Thank you very much for your professional suggestion, actually, we have used both the 1-3 NOEs (H-3/H2-5 and H-4/H2-25, Line 194) and heteronuclear coupling constant (J H-3/H-4= 0 Hz, Line 195), instead of NOE between vicinal protons such as H-3/H-4, to figure out the relative configurations of compound 6, just as you suggested here. And the quality of Figure 7 has been revised to prevent misleading readers.

Reviewer 3 Report

Manuscript ID: marinedrugs-404835. The structure elucidation seems to be correct, therefore, the submitted manuscript can be accepted. Nevertheless, the following changes should be done before publication:

1. The 1H and 13C NMR spectra presented in Supporting Information contain considerable number of signals in the range of 0-2.5 ppm for 1H NMR and 0-40 ppm for 13C NMR. It is difficult to read and prove the correctness of structures. Please make zooms in these regions in Figures S9, S10, S13, S14, S21, S22, S28, S29, S32, S33, S36, S37, S44, and S45.

2. Due to the lack of scale on the axes of 2D NMR spectra, it is also difficult to read them. Please, mark at least critical correlations described in the main text.

3. The structure of compounds 8-10 are known from literature. It is OK. However, they are discussed in the text, and thus, at least the 1H and 13C NMR spectra must be given in Supporting Information.

4. In Figure S29 some signals are not marked. According to Table 2: No 5, 15, 16, 19, 20.

5. Figure 1. There is lack of numeration for the compounds 5a/5b. I know that this is a detail, however, the person who reads should easily determine the numbering of cyclohexene ring.

Author Response

1. The 1H and 13C NMR spectra presented in Supporting Information contain considerable number of signals in the range of 0-2.5 ppm for 1H NMR and 0-40 ppm for 13C NMR. It is difficult to read and prove the correctness of structures. Please make zooms in these regions in Figures S9, S10, S13, S14, S21, S22, S28, S29, S32, S33, S36, S37, S44, and S45.

Zooms have been added in all of the 1D NMR spectra. 

2. Due to the lack of scale on the axes of 2D NMR spectra, it is also difficult to read them. Please, mark at least critical correlations described in the main text.

Key 2D NMR correlations for the new constructed compounds have been described in the main text, and some of the 2D NMR spectra have been raised of their quality.

3. The structure of compounds 8-10 are known from literature. It is OK. However, they are discussed in the text, and thus, at least the 1H and 13C NMR spectra must be given in Supporting Information.

The 1H and 13C NMR spectra for these known compounds have been added into Supporting Information.

4. In Figure S29 some signals are not marked. According to Table 2: No 5, 15, 16, 19, 20.

We have checked Figure S29, and the reason why these peaks were unmarked is there are too many chemical shift values presented in such a limited area. We have cut the spectrum into two separate pictures and the problem has been solved.

5. Figure 1. There is lack of numeration for the compounds 5a/5b. I know that this is a detail, however, the person who reads should easily determine the numbering of cyclohexene ring.

The numeration for the compounds 5a/5b has been added into Figure 1.